# Korean Nursing Students’ Experiences of Virtual Simulation Programs Replacing In-Person Mental Health Nursing Practicum during the COVID-19 Pandemic

**DOI:** 10.3390/healthcare12060685

**Published:** 2024-03-19

**Authors:** Sunyoung Lee, Eunyoung Park, Hyun-E Yeom

**Affiliations:** College of Nursing, Chungnam National University, Daejeon 35015, Republic of Korea; leesun033@hanmail.net (S.L.); yeom@cnu.ac.kr (H.-E.Y.)

**Keywords:** COVID-19, mental health nursing practicum, nursing education, nursing student, qualitative research, simulation training

## Abstract

This qualitative study explored the experiences of nursing students whose clinical practice in mental health nursing had been substituted with virtual simulation programs due to the COVID-19 pandemic. The participants were ten nursing students who had undergone a virtual simulation program-centered practice, replacing the traditional clinical practice in mental health nursing and previous clinical practice in mental health nursing. The data were collected through in-depth individual interviews from January to February 2021. Following Braun and Clarke’s method, the thematic analysis identified five themes and ten sub-themes. The five themes included the following: (1) lack of vibrancy in the actual clinical setting, (2) limited direct and indirect practical experience, (3) performing diverse roles in a virtual setting, (4) learner-directed practicum, and (5) sense of relief due to a safe virtual practicum environment. The participants recognized the limitations of the practice, particularly regarding communication with patients with mental disorders in the virtual simulation program. However, their perception of nursing underwent a positive change through the indirect clinical practice experience. Accordingly, it is necessary to develop a platform for the mental health nursing practicum that can easily interact with clients and to establish a hybrid practice that combines the clinical practice and virtual simulation practice.

## 1. Introduction

Nursing is an applied discipline that integrates clinical practicum with theoretical knowledge, empowering students with the essential skills, knowledge, and attitudes necessary for developing a professional identity as a nurse [1]. Clinical practicum is a critical component of nursing education that enables students to experience socialization in a clinical setting and acquire the core values, behaviors, and attitudes of the nursing profession [2]. Additionally, clinical practicum offers nursing students the opportunity to apply theoretical knowledge to develop the necessary skills and competencies in a clinical setting that enhance their learning motivation [3].

### 1.1. Study Rationale 

Following the outbreak and escalation of the Coronavirus Disease 2019 (COVID-19) in March 2020, the World Health Organization (WHO) declared the situation a pandemic, the highest level of warning for infectious diseases [4]. As the unprecedented health crisis had disrupted the clinical practicum training, the Korean Accreditation Board of Nursing (KABON) issued guidelines for conducting on-campus practicum courses as an alternative to the in-person practicum [5]. Based on the government’s guidelines, nursing schools adopted various measures for education, infection control, and conditions at the clinical practicum sites to find a substitute for their clinical practicum [6,7], including virtual simulation programs, simulation model practicums, nursing skills videos, and case presentations via live video conferencing [8]. Mental health nursing practicums are designed to provide nursing students the competence to develop positive therapeutic relationships with people experiencing mental disorders through communications training [9]. However, during the COVID-19 pandemic, psychiatric care facilities restricted nursing students’ access to the psychiatric wards due to the infection-prone contamination on these floors [10]. As a result, virtual simulation programs were adopted to replace the mental health nursing practicum.

Virtual simulation programs replicate clinical environments in a computer, allowing participants to communicate, make decisions, and perform nursing activities with virtual patient scenarios [11]. These programs offer the advantage of allowing multiple students to participate simultaneously without the constraints of location and time; because of this, they are widely used in medical and nursing practicum education [12]. Examples of virtual simulation programs include vSim (Virtual Simulation for Nursing; vSim^®^) and Electronic Medical Record (EMR; Bit Nix Academy™). vSim, co-developed by Wolters Kluwer and Laerdal Medical, is designed to run nursing scenarios in a clinical setting, enabling learners to interact with patients in a virtual environment similar to a clinical setting [13]. The EMR system generates and manages medical records, including diagnoses, nursing interventions, medical histories, medications, and tests performed, where students can explore patient data and improve their competencies to engage in the nursing process [14].

Previous studies have shown that virtual simulation education enhances learners’ self-efficacy, motivation, nursing skills proficiency, confidence, and theoretical knowledge, thus increasing their satisfaction with the practicum [13,14,15,16,17,18,19]. However, the fact that most virtual simulation programs are produced in English presents language barriers, and the inability to meet actual patients in clinical settings limits the provision of nursing care; additionally, the restricted information and isolated learning environments that occur in virtual settings are found to induce anxiety and depression [11,15]. Therefore, it is important to investigate the experiences of nursing students’ (as they are the primary subjects of education) in virtual simulation programs to understand the limitations of replicating clinical practice settings in virtual simulations [13]. Examining virtual simulation program practices can provide direction for developing strategies that are tailored to the unique needs of nursing students in Korea [18]. Building therapeutic relationships with clients is a critical aspect of the mental health nursing practicum; unfortunately, studies exploring nursing students’ educational experiences in a mental health nursing practicum in a virtual environment are limited [10]. Therefore, it is imperative to understand in depth the nursing students’ perceptions and experiences with virtual simulation programs in mental health nursing. The findings would be useful for providing high-quality alternative mental health practicums in the event of restrictions caused by another infectious disease outbreak. For this purpose, qualitative research is beneficial in understanding and describing the experiences and perspectives of students in virtual simulation programs as an alternative to a clinical practicum [20]. 

### 1.2. Aim

This study aims to explore nursing students’ experiences in a virtual simulation practice as an alternative to a mental health clinical practicum during the COVID-19 pandemic, leading to the key study question: What are the experiences of nursing students with virtual stimulation programs as an alternative to mental health nursing practicums? 

## 2. Materials and Methods

### 2.1. Study Design

This qualitative study aims to explore the experiences of nursing students using virtual simulation programs, specifically as an alternative to an in-person mental health nursing practicum during the COVID-19 pandemic. This study was conducted in adherence with the consolidated criteria for reporting qualitative research (COREQ) checklist [21]. 

### 2.2. Study Participants

The purposive sampling method was used to select participants from a pool of nursing students who had experienced both an in-person mental health nursing practicum prior to the COVID-19 pandemic and virtual simulation programs following the outbreak of the COVID-19 pandemic. Participants were selected after they expressed voluntary interest in response to the recruitment announcement and were found to meet the aforementioned selection criteria. From among the 80 fourth-year nursing students of South Korea’s J University in C Province who had completed a two-week (nine hours per day for a total of ten days) virtual simulation program-based practicum for the course “Mental Health Nursing Practicum 2” in the first semester of 2020 and had prior experience with the in-person mental health nursing practicum before the outbreak of the COVID-19 pandemic, ten students were enrolled. Their eligibility was verified, and after obtaining their informed consent, these ten participants (eight women and two men) were interviewed until data saturation was reached. None of the participants withdrew their consent or discontinued study participation. The mean age was 24.5 years (Table 1).

### 2.3. Virtual Simulation-Based Practicum versus In-Person Mental Health Nursing Practicum 

The virtual simulation-based practicum was designed and implemented over two weeks for a total of ten days, based on the existing clinical practicum standard of nine hours per day. To enhance the ability to set nursing priorities and clinical reasoning skills [22], vSim was used. vSim activities include a pre-quiz, simulation exercises, post-quiz, debriefing, and other assignments, consistent with the given scenarios [13]. Additionally, the program utilized the learning management system (LMS)—an essential tool for supporting learning in an online environment and facilitating interactions between instructors and students [23]—and Mosby’s Nursing Skills (NSL), which comprises a 32-item nursing skills framework that was developed by Mosby and translated into Korean under the supervision of the Korean Society of Nursing Science [23,24,25]. The LMS played a central role in managing the offline curricula, including learner grades, progress tracking, report submission, and attendance, by transitioning them into an online space dedicated to communication and collaboration between the instructors and learners [23]. The evaluation occurred upon completion of the NSL’s psychiatric nursing skills video, which presented evidence-based nursing skills tailored for each subject and aligned with the Korean clinical practice. These materials represent the latest advancements in clinical skills training [25].

In the virtual simulation-centered mental health nursing practicum, the ten students aimed to meet the daily learning goals by completing content on mental health nursing skills based on NSL and engaging in ten mental health nursing scenarios in vSim per the protocol. The students continued their learning until they reached a 90% completion rate, at which point they captured and submitted a screenshot of the score, indicating this achievement to the LMS. Subsequently, group debriefing sessions were held, where the students presented their application of the nursing process to scenarios in vSim, and the faculty provided feedback and additional instructions to enhance the participating students’ practical competencies (Figure 1).

BPD = borderline personality disorder; LMS = learning management system; NSL = nursing skills (a learning management system platform integrated with a nursing skills database and online educational function by ELSEVIER, Korea) learning; PTSD = post-traumatic stress disorder; vSim = virtual simulation for nursing.

### 2.4. Data Collection 

The data were collected via individual interviews from nursing students who voluntarily provided written informed consent over a one-month period from 26 January to 9 March 2021. The first author (SL) posted the study recruitment posters in the group chatroom for students at the targeted study facility and on the bulletin board at the departmental office of the facility. The students who expressed a willingness to participate in the study were contacted individually to eliminate any direct or indirect influence of the faculty and to ensure their voluntary participation. The first author was a woman serving as a teaching assistant in the Nursing Department at J University who had established a rapport with the students. Each participant was interviewed between two and three times, and each session lasted about 30 to 60 min. In compliance with the COVID-19 infection prevention guidelines, consent forms were signed in the author’s office, and interviews were conducted over the phone at times convenient for the participants to minimize face-to-face contact. To capture nuances such as intonation, laughter, and hesitation, notes were taken during the interviews, and with the participants’ consent, all the sessions were recorded using the voice recording feature on the first author’s (SL) personal cellphone. Within 24 h of an interview, the first author (SL) transcribed the participants’ verbal expressions during the interview ad verbatim. Follow-up interviews were conducted with four participants, and the additional interviews were conducted via phone and text messages due to geographical distance. The data collection and analysis proceeded concurrently until no new concepts or significant data emerged; the first author shared the interview transcripts with two other authors (EP and HY), and the point of saturation was determined after an independent review and meetings. Open-ended questions were used for the interviews to create a comfortable atmosphere, encouraging the participants to freely express their thoughts and opinions on their experiences. After concluding the interviews, the analysis of the interview contents was shared with all ten participants to ensure it accurately reflected their experiences and perspectives. The key interview question was as follows: Please tell me about your experience with the virtual simulation program as a replacement for the mental health nursing practicum due to COVID-19. This was followed by more specific questions while encouraging the participants to express their feelings and experiences, ensuring to capture participants’ perspectives and thoughts on the virtual simulation-centered practicum in mental health nursing without distorting them (Table 2).

### 2.5. Data Analysis

The data analyzed in this study comprised 67 pages of transcribed A4-sized documents and field notes. The data were analyzed consistent with the six phases of thematic analysis proposed by Braun and Clarke [26] to describe the meanings of common themes within the data (Figure 2).

### 2.6. Ethical Considerations 

This study was approved by the Institutional Review Board prior to the data collection (JIRB-2020123101-01-210118). To ensure the ethical consideration of the participants, the aim, methods, and procedures of the study as well as the assurance of voluntary participation, confidentiality, anonymity, and right to withdraw without penalty were explained before the interviews, and the participants signed a written informed consent form. The recruitment of participants was conducted by the first author (SL), who was not involved in teaching or grading the virtual simulation-based mental health nursing practicum after students’ grades for the course had been submitted to protect vulnerable participants and ensure grading impartiality. As a token of appreciation for their participation, the participants were provided mobile gift vouchers after the interviews.

### 2.7. Preparation for Researchers and Measures to Ensure Rigor of Study 

The first author (SL) had completed a qualitative research methodology course during a doctoral program and attended qualitative research conferences to enhance her skills. Prior to this study, she had engaged in qualitative research based on interviews with students. The corresponding author (EP) is an experienced qualitative researcher engaged in teaching qualitative research methodology courses and has consistently conducted research related to nursing education. The third author (HY) is a researcher with over ten years of educational experience in nursing.

The rigor of this qualitative study was ensured by complying with the four-dimensional criteria proposed by Sandelowski [27]: credibility, auditability, confirmability, and fittingness. To ensure credibility, interviews were conducted in a positive and motivational atmosphere using participants’ personal phones, and the interview contents were transcribed immediately after the interviews and read repeatedly over a long period to interpret the participants’ experiences as stated. Additionally, the participants were asked to verify the analysis results for accuracy and intent. For auditability, recorders and field notes were used during the participant interviews to prevent data omission and increase the accuracy of the analysis. The data analysis was mutually reviewed and discussed by all three authors (SL, EP, and HY) to ensure that the results of the analysis were not distorted. Consistency was maintained by seeking advice on the study process and analysis results from an external professor experienced in qualitative research. Confirmability was maintained by noting researchers’ biases during the data analysis process to ensure that the analysis results did not reflect the researchers’ judgments and biases. A colleague was consulted for advice and for the evaluation of the data analysis and interpretation to prevent errors and biases and to maintain the objectivity by discovering implications that the authors may have missed. To ensure fittingness, adequate data were collected until saturation, and the phenomenon was described as concretely, vividly, and richly as possible. Furthermore, three nursing students with experience in virtual simulation-based mental health nursing practicums who were not participating in this study were consulted to ensure fittingness.

## 3. Results

During the COVID-19 pandemic, nursing students’ experiences with virtual simulation versus the in-person mental health nursing practicum emerged in the form of five themes and ten sub-themes (Figure 3). The five themes were as follows: (i) lack of vibrancy in the actual clinical setting, (ii) limited direct and indirect practical experience, (iii) performing diverse roles in a virtual setting, (iv) learner-directed practicum, and (v) sense of relief due to a safe virtual practicum environment. 

### 3.1. Lack of Vibrancy in the Actual Clinical Setting

The participants were confused and perceived limitations in the virtual simulation practicum compared to their previous clinical experience of working in psychiatric units, as it was difficult to identify the symptoms of different mental disorders among virtual patients. They stated that nursing interventions in the mental health nursing practicum must be implemented after building rapport with the patient, and the virtual patient expressed emotions verbally and lacked non-verbal behaviors, making them feel like they were interacting more with machines than humans. The participants also stated that it was difficult to differentiate between patients’ symptoms and conditions in the virtual simulation program, as only limited information was available. They struggled to establish nursing priorities as they focused on analyzing scenarios by applying theoretical knowledge, inferring, and hypothesizing. They stated that the practicum was limited by the lack of vibrancy in the virtual clinical setting. Thus, this theme comprised two sub-themes: (a) facing limitations in therapeutic communication and (b) confusion in distinguishing symptoms of mental disorders. 

#### 3.1.1. Facing Limitations in Therapeutic Communication

The participants stated that the mental health nursing practicum requires them to observe the client and serve as a therapeutic tool to engage in communications involving reflecting, clarifying, and emphasizing. However, they faced challenges during the virtual simulation, as patients expressed emotions solely through short verbal responses such as “sad” or “depressed”, with limited non-verbal behaviors that are crucial for understanding and interpreting these expressions. The structured format of responding to pre-set questions in the virtual simulation program and patients’ monotonous responses hindered interactive communication. As a result, the study participants had practical difficulties in building therapeutic relationships with patients, which in turn hampered the accurate assessment of patients’ nursing needs.

In mental health nursing practicums, communication is essential… (omitted) vSim made communication challenging in mental health nursing. Even with similar questions, I was like, what is the difference here? You’re supposed to judge based on the person’s facial expression, but the patient is just there, with eyes blinking. So, I couldn’t tell if the patient was in a good mood or irritable. (Participant 4).

Should I call it the difference in vibrancy? I’ve been to a psychiatric hospital. There, you could see everything. People were running around, even fighting. However, in the vSim program, it’s just words like ‘sad’, so it wasn’t realistic in terms of the psychiatric aspect. It just felt like a machine. (Participant 5).

#### 3.1.2. Confusion in Distinguishing Symptoms of Mental Disorders 

The participants stated that in their classes, they studied that manifestations and behaviors could vary widely even with the same mental disorder. However, they stated that patients with mental disorders in the virtual simulation program presented different behaviors and speech patterns from the ones they had observed and experienced in their previous psychiatric unit rotations. Participants found it challenging to discern the typical characteristics of different mental disorders, such as schizophrenia, bipolar disorder, and depression, due to the monotonous verbal and non-verbal symptoms presented in the simulation. They were instructed to address the mental symptoms presented in the scenario based on their theoretical knowledge; however, they were confused when they attempted to distinguish the symptoms due to the vague presentation of verbal and non-verbal behaviors in the simulation scenarios. Moreover, managing virtual patients felt mechanical due to the lack of vibrancy in the virtual simulation setting. 

In the previous rotations, I felt that all the 100 schizo (schizophrenia) patients I see are different, but (in the virtual simulation), it felt like that all the schizo patients, including those with bipolar disorder and depression were all similarly programmed. I think this is a limitation of the virtual simulation. (Omitted) Even if two patients have the same disease, their symptoms differ substantially and the way they behave is also different. In vSim, they all seem similar … For example, schizo patients have the same symptoms and say the same things and that really felt different from my clinical practice. (Participant 2).

In the virtual simulation, there were only two answer options in the protocol. When the patient says, ‘I don’t know where I am’, it gives me two choices ‘This is a hospital. Wake up’! (laughs). Come on, really? ‘Wake up’ is so inappropriate. (Participant 3). 

### 3.2. Limited Direct and Indirect Practical Experience 

The participants were disappointed by the limited opportunities for in-person mental health rotations due to the COVID-19 pandemic. They voiced their concerns about the difficulties they could face while adapting to the clinical settings and had a vague sense of anxiety about the future due to the inadequate training through clinical rotations as prospective nurses. Thus, this theme comprised two sub-themes: (a) limited learning experiences during an infectious disease public health crisis and (b) anxiety about actual clinical competence.

#### 3.2.1. Limited Learning Experiences during an Infectious Disease Public Health Crisis

The participants acknowledged the need to restrict clinical practicums in psychiatric units and public health centers due to the risk of contracting COVID-19. However, they felt that experiencing the unique conditions of a pandemic is both their right and duty as future healthcare professionals. Even while understanding the inevitability of these restrictions for the safety of the learners and patients, the participants were frustrated by missed clinical education opportunities, as they could not gain direct exposure to field guidelines or response strategies employed in the practice. 

In my third year, I saw a schizophrenia patient during my mental health rotations, and I was supposed to see patients with alcoholism in the first semester of my fourth year, but I was sad that I couldn’t experience this because of the COVID-19 situation. I was curious about why alcoholics drink so much. I felt like it was unfair because I couldn’t learn what I should’ve because of the pandemic. (Participant 4).

#### 3.2.2. Anxiety about Actual Clinical Competence

The participants expressed heightened anxieties as they missed their clinical rotations regarding their impending graduation and transition into professional nursing roles. They questioned themselves, reflecting “Will I be able to do well once I start working?” “Will I be able to do as well as before?” They voiced concerns regarding their abilities to address emergencies in real-life clinical settings, as clinical competence is enhanced during in-person clinical rotations while interacting with patients, even if they are primarily observation-based, because virtual simulations have constraints in collecting patients’ subjective and objective symptoms.

Now, I definitely have a different mindset compared to my third year. Graduation is approaching, and now is when the way I view and observe teachers’ changes. I just felt a little anxious that I couldn’t do my rotations when I really needed them. (Participant 10).

I haven’t actually seen or done this in a hospital, and models are so different from real people. I wondered if I would be able to do well in real clinical settings. (Participant 10).

### 3.3. Performing Diverse Roles in a Virtual Setting

The participants noted that they had limited opportunities to actually perform nursing activities during their rotations; however, the virtual simulation space allowed them to fully care for a patient from admission to discharge as the primary nurse rather than a student nurse, enabling them to engage in various roles that were restricted in an in-person clinical practicum. While they did not have much time to collect patient data during their in-person rotations, vSim allowed them to access patient data as needed, and this helped them make better evidence-based nursing diagnoses. They reported that the simulation strengthened their perception of nursing as a profession and not just as an occupation involving repetitive tasks. Thus, the theme of performing diverse roles in a virtual setting comprised three sub-themes: (a) indirect experience as a prospective nurse, not a student nurse, (b) strengthened awareness about the nursing profession, and (c) developing critical thinking skills. 

#### 3.3.1. Indirect Experience as a Prospective Nurse, Not a Student Nurse

The participants stated that their experiences during in-person clinical rotations are limited to communications or observations, but virtual simulation allowed them to engage in more expanded roles that are not permitted in real-life settings as a student. In the vSim scenarios, they communicated with patients with mental disorders, reported their status to physicians, and administered the prescribed oral medications, injections, and tests. Such indirect experience as practicing staff provided them with a sense of accomplishment. They mentioned that this experience provided them an opportunity to gain knowledge on the various tests and medications used for the symptoms of mental disorders. 

What you can do is limited as a student nurse; however, I was able to perform various roles in the virtual simulation as a student nurse. (Omitted) In virtual simulation, you can experience several things that are not allowed in an actual clinical setting, at least indirectly. (Participant 9).

In the vSim, we do everything as nurses; therefore, taking care of a patient completely, that is, beyond what you’re allowed to do as a student, by clicking on features is the greatest change. It was helpful because I was able to learn and understand that you should act like this in this case. It was good that I was able to do things that I wasn’t allowed to in hospitals and gained an indirect experience with things like how to address these, as a prospective nurse. (Participant 4).

#### 3.3.2. Strengthened Awareness of the Nursing Profession

During clinical rotations, the predominant perception among the study participants was that nurses are workers who engage in repetitive tasks, as opposed to professionals, but as they are engaged in the diverse roles of a nurse through virtual simulation, they experienced the importance of timely and accurate nursing care in a hectic environment. They were able to truly feel what nursing is about. One participant stated that their professional nursing identity was bolstered, as they developed a sense of duty that nursing is a noble profession that takes responsibility for human lives. 

I think nurses are professionals, and that this is true nursing. In clinical practice, I had felt that, yeah, of course they are professionals, but I felt they do more routine work. I had a stronger perception about nursing being routine work than being a profession. However, vSim shows abstractly that nursing is a profession. (Participant 5).

#### 3.3.3. Developing Critical Thinking Skills

The participants noted that the EMR system in the virtual simulation scenario paralleled real hospital EMRs, and repeatedly obtaining patients’ vital signs, medical and family histories, prescriptions, and mental health assessments improved their thinking abilities. They believed that the EMR practice would help them adapt to actual hospital EMRs. While they were allowed to access EMRs during in-person clinical rotations, they did not feel comfortable accessing them without ample time due to the ongoing busy work activities with their nurse preceptor. However, they were able to focus on the analysis without being conscious of other people in the virtual simulation, which helped them foster skills to obtain patient information in a timely manner and manage patient cases. 

Analyzing EMR… One thing I really appreciated was learning how to analyze quickly. You know, analyzing patients. In an actual hospital, it’s hard to take your time because everyone is so busy, but I liked the EMR part (in the virtual simulation). Even though EMR systems differ at each hospital, they all have records, such as patient medical history. It was super helpful because I was able to learn how to read them and understand them quickly. (Participant 6).

Eventually, you must learn how to use an EMR, and understand that this was a learning opportunity. It’s not identical, but I was able to learn about certain features of EMR and realize that in-depth information is incredibly helpful, especially for conducting case studies. (Participant 1).

### 3.4. Learner-Directed Practicum

The participants highlighted a distinct shift in their practicum experience with the virtual simulation programs compared to traditional clinical settings. While the clinical practicum often involved cautious and reserved interactions with real patients due to their underdeveloped communication skills, the virtual environment provided a less intimidating space for engaging with simulated patients. This allowed learners to assume responsibility for their learning experience, actively diving into the practicum without the pressure of direct patient contact. Observing the progression or deterioration of a patient’s condition in the virtual simulation helped expand their nursing competencies. Additionally, completing daily assignments and following vSim scenario protocols facilitated a deeper understanding of the disease pathophysiology and medication indications, enriching their ability to develop nursing care plans. This theme consisted of two sub-themes: (a) gaining a sense of accomplishment through active participation and (b) self-directed learning.

#### 3.4.1. Gaining a Sense of Accomplishment through Active Participation 

The participants noted that their interactions with patients with mental disorders were mostly observational in actual clinical settings, with direct nursing actions taken cautiously for fear of negatively impacting the patient. However, virtual environments in the vSim program alleviated this pressure, enabling them to participate more actively, eventually increasing their confidence and sense of achievement. After completing nursing actions within the vSim scenarios, they were given immediate feedback on their performance, and this helped them review the errors that were made during the simulation practicum.

In the actual clinical practicum, I was mostly observing, and I was very cautious about performing any procedures on patients because I was afraid of harming them. However, with vSim, since it’s virtual, I could be more proactive, and it helped build my confidence. (Participant 1). 

Based on patient symptoms, I was able to report to doctors and administer the prescribed medications. Since it’s online, I could check the outcomes of my interventions. There are more opportunities in the virtual simulation practicum and passing the scenarios during the first try gave me a sense of achievement. (Participant 2).

#### 3.4.2. Self-Directed Learning

The participants reflected that during clinical practicum, they would resolve patient queries by consulting with other nurses on the floor, whereas virtual simulation offered an environment that focused more on individual students, where they could direct their own searches to resolve the immediate problems. The participants were able to utilize their time autonomously and flexibly, and they mentioned that setting clear learning objectives and supplementing theoretical knowledge independently through self-directed learning was a rigor of the program.

vSim provided ample time for self-study, which was great for filling in the gaps in my theoretical knowledge. The ability to manage my time was probably the biggest advantage. (Participant 1). 

During clinical practicum, I sometimes had to skip questions because I had no time, but in virtual simulation, I felt I could search for anything that I was curious about. (Participant 3). 

The virtual simulation practicum allowed me to fully concentrate on studying… I could focus on disease cases. There was plenty of time to study. I could explore more books and even research papers online. (Participant 8).

### 3.5. Sense of Relief Due to a Safe Virtual Practicum Environment

The participants acknowledged that they had experienced a sense of relief and safety when virtual simulations replaced clinical practicums, alleviating fears of COVID-19 transmission in hospital settings. Additionally, training during in-person clinical rotations is dependent on schedules that are based on the statuses of various hospital floors, whereas virtual simulations offered the flexibility to use the practicum time more adaptively, as there were no time constraints. Thus, this theme comprised the sub-theme of feeling relaxed and safe in a virtual clinical practicum setting. 

#### Feeling Relaxed and Safe in a Virtual Clinical Practicum Setting

The participants stated that contracting an infection is always a risk in the clinical setting despite following all infection prevention protocols, such as social distancing, mask-wearing, and handwashing. However, they felt comparatively safer from the risk of COVID-19 in virtual simulations, as these simulations were run in less crowded settings. In-person mental health nursing rotations allowed the participants to observe actual patients based on theoretical learning, whereas virtual simulations offered more detailed feedback and interaction from instructors, enhancing the integration of theory and practice. Some of the participants mentioned that their previous experiences with in-person mental health nursing rotations induced elevated levels of anxiety due to fear of patients. In contrast, the virtual simulation program provided a physically and psychologically more relaxed learning environment.

Well, hospitals are replete with the risk of infection. Even with masks and hand sanitization, I was really worried that I might get COVID. However, in virtual simulation, there are fewer people and only students around, so it felt safer. (Participant 7).

Viewing patients during my previous mental health nursing rotations was intriguing, but it didn’t differ much from my classroom learning. The professor actually provided more detailed explanations, so I really liked how it supplemented my theoretical understanding. (Participant 2).

After my mental health nursing rotation in the third year, I had nightmares for two weeks. It was super tough. A patient asked if I had eaten, so I replied that I had dined. However, the patient would repeat the question again. I was like, oh, alright, this is a psychiatric floor. In the virtual simulation, there were no such scary elements, so I felt more relaxed. (Participant 9). 

## 4. Discussion

This study aimed to explore and describe nursing students’ experiences with virtual simulation programs, specifically as an alternative to the mental health nursing practicum during the COVID-19 pandemic, using Braun and Clark’s [26] thematic analysis. Based on our analysis, we identified ten sub-themes categorized under five themes: (i) lack of vibrancy in the actual clinical setting, (ii) limited direct and indirect practical experience, (iii) performing diverse roles in a virtual setting, (iv) learner-directed practicum, and (v) sense of relief due to a safe virtual practicum environment.

First, under the theme “lack of vibrancy in the actual clinical setting”, participants noted challenges in distinguishing patients’ symptoms in a virtual simulation practicum compared to an in-person clinical practicum, attributable to the limitations in patient interactions and communications. This finding is similar to the previous ones, suggesting that one of the learning objectives of mental health nursing practicums is acquiring communication skills to build therapeutic relationships through patient observation and interviews [28], which is compromised by the diminished vibrancy of virtual settings, causing confusion among students [29]. This may be attributable to the limited responses and interactive dialogue in the virtual simulation program that hamper interactions, as well as the need to choose only a limited number of nursing activities that must be completed within the program. To address these issues, virtual reality-based practicums may be considered in situations that are restricted in clinical practicums, such as an infectious disease public health crisis. Previous findings should be considered, wherein students who engaged in a virtual reality-based mental health nursing practicum that vividly resembled the clinical setting in terms of communication and interactions developed an increased confidence in therapeutic communication with clients [28]. However, rather than solely focusing on skill-based programs, it is crucial to contemplate the psychiatric field specifically. The care and humanization of individuals with mental disorders must incorporate comprehensive insights into the psychiatric clinical practice field. Therefore, when considering the implementation of a mental health nursing practice based on virtual reality, actively designing the practice using standardized patients and role-plays can enhance therapeutic communication, interaction, and teamwork: essential components in the field of mental health nursing. This approach aims to bridge the gap between virtual reality and the reality of the clinical field, which may be lacking in certain aspects of mental health representation.

Second, under the theme “limited direct and indirect practical experience”, the participants expressed relief from the risk of contracting a COVID-19 infection and regret for not experiencing actual clinical settings during the pandemic. This illustrates previous findings, that nursing students who underwent clinical practicums during the pandemic experienced psychological distress and anxiety due to concerns about the COVID-19 infection [12]. This is also consistent with previous results, that a shift to online practicums limited the clinical rotation opportunities in several units and caused anxiety and depression [30]. The participants in this study were fourth-year students nearing graduation, and they mentioned that failing to fully establish clinical competencies during their final clinical rotations diminished their confidence as prospective nurses. For prospective nurses, high levels of confidence and satisfaction are crucial for developing a positive perception of the nursing profession [29]. While the pandemic introduced unpredictability and discontinuity in clinical practicums, it also highlighted the significance of nursing [31]. Adjusting to the “new normal” since the onset of the COVID-19 pandemic has prompted a re-evaluation of the teaching approaches for nursing practicums [29]. Compared to other areas, students in mental health nursing practicums frequently experienced negative attitudes, fear, and anxiety, which impaired their abilities to form therapeutic relationships and presented challenges during the practicum [13]. As an alternative, role-plays or standardized patients can be used to enhance communication skills and interview techniques that are required in mental health nursing practicums, specifically during times when clinical practicum experiences are limited. These approaches could reduce students’ tensions and anxieties while boosting their confidence in clinical practice [10,32,33,34]. Therefore, role-play clinical simulations should be considered to develop effective coping strategies, communication skills, and teamwork skills among learners. 

Third, under the theme “performing diverse roles in a virtual setting”, the participants highlighted that the virtual simulation enabled them to gain an indirect experience in various nursing roles that are not typically accessible to students during clinical rotations, and that these experiences strengthened their perceptions of the nursing profession and fostered their critical thinking skills. Supporting our results, previous findings show that students’ prioritization and critical thinking skills improved with exposure to various cases as proactive nurses [8,11,15,19]. The participants reported that the virtual simulation program provided indirect experiences with various psychiatric patients and allowed for the analysis of virtual EMR scenarios, such as those in actual hospital settings, as well as helped to cultivate their critical thinking skills. Compared to the observation-focused training with a traditional in-person clinical practicum, gaining indirect experiences of the clinical practice through a virtual simulation positively transformed their perceptions of the nursing profession. The professional identity in nursing is shaped through theoretical education and a clinical practicum and grows with clinical practice, and the first clinical rotation has a positive effect on the perception of the image of a nurse and the formation of a desirable nursing professional identity [35]. The findings of this study confirmed that indirect experiences with various nursing tasks through a virtual simulation strengthened students’ perceptions of the nursing profession. Forming a desirable nursing professional identity contributes to advancements in the nursing practice and patient safety and facilitates interdisciplinary interactions [36]. 

Fourth, under the theme “learner-directed practicum”, participants became more actively involved in practicums by performing nursing activities autonomously and independently. Such self-directed learning helped them achieve their learning objectives, in turn, increasing their sense of accomplishment. Our results are supported by previous findings, that the flexibility and autonomy offered by virtual simulation practicums allowed for active involvement and repeat learning, increasing satisfaction with the practicum [5,17], and nursing students participating in online classes during the COVID-19 situation showed improved an academic performance and self-efficacy as a result of the flexible learning environment and learner-centered practicum [1,37]. During online learning, participants in this study engaged actively without fear of harming the patient; moreover, repeat learning, enabled by the lack of time and spatial constraints, allows for self-directed learning and boosts the learning motivation [5,7,37]. The participants in this study were able to focus on repeat learning through the virtual simulation without being burdened by the concern of harming the patient, and they were able to correct the errors in their nursing interventions, thereby facilitating learning. Immediate feedback from the virtual simulation program and assignments with learning outcome standards set by instructors facilitated the students’ self-directed learning. Simulation-based education is considered an effective approach, as it enables repeat learning on specific cases, and thus, positively affects clinical outcomes and patient care and improves academic achievement among learners with social phobia [37,38]. Thus, virtual simulation programs can enhance the sense of achievement for learners experiencing their first mental health nursing practicum or those apprehensive about mental illness, potentially leading to a positive shift in perceptions toward patients with mental disorders.

Fifth, under the theme “sense of relief due to a safe virtual practicum environment”, participants experienced physical and psychological comfort during the virtual practicum, attributable to the mitigated concerns about the transmission of the COVID-19 infection in hospitals. This aligns with previous studies where a virtual practicum setting helped students feel safe from both patients and nurses and experience comfort during the practicum program [13]. Our virtual simulation practicum occurred in the early part of the COVID-19 outbreak, when there was tremendous fear and anxiety about the infection following WHO’s declaration of a pandemic and Korea’s red-level alert [4]. Thus, training in a virtual setting instead of a regular hospital setting may have provided psychological comfort. Furthermore, the instructors played a facilitative role by providing ample feedback and encouraging active participation in the practicum. The report that students preferred debriefing with colleagues and receiving feedback from instructors following a virtual simulation supports our results [13]. In addition, the virtual simulation practicum created a psychologically comfortable environment, enhancing learner engagement more than traditional clinical practicums [11]. Nursing students experiencing their initial psychiatric nursing clinical practicum often encountered prejudice, fear, and anxiety about mental illness [39]. However, the study participants demonstrated reduced levels of uncertainty, fear, and anxiety during their second psychiatric nursing practice in the virtual simulation setting. The possible reason may be the emotional aspect of the Korean culture, particularly the heightened sensitivity to the risk of infection, resulting in a positive response and a sense of relief from COVID-19. As students feel safe and relaxed in virtual simulation practicums during periods of chronic anxiety about contracting infectious diseases, instructors should develop various teaching methods for virtual simulation practicums to maximize the positive impact of virtual simulations that facilitate student participation through immediate and frequent feedback [40].

This study has certain limitations. First, considering the transferability of qualitative research, caution is needed when applying the findings to other educational curriculums or other practicum courses. Second, vSim, which utilizes virtual reality, is a web-based, self-directed learning program that was implemented in this study as a non-face-to-face practice program. Caution must be exercised when interpreting the results to avoid an overly generalized application of vSim as representative of the entire spectrum of virtual reality programs. Additionally, as we relied on the participants’ recalls of their past clinical practicum experiences to collect the data, other data collection methods, including observation, were not considered to enrich the study data.

## 5. Conclusions

This study is significant, as it provides understanding and illustrates nursing students’ experiences with virtual simulation during the COVID-19 pandemic. Nursing students encountered the limitations of therapeutic communication with patients with mental disorders due to the lack of vitality in the virtual simulation programs, but their sense of accomplishment improved through independent nursing performance and repeated learning led by learners in a safe virtual practicum environment. The findings emphasize the practice as an alternative to a mental health nursing practicum and provide foundational data for developing effective educational strategies and interventions for mental health nursing practicums in the event of a new infectious disease pandemic. Based on the results, this study makes the following recommendations: First, studies should develop a platform for a mental health nursing practicum that incorporates the unique culture of mental health nursing and facilitates the learning of therapeutic communication skills. Second, hybrid programs that combine in-person clinical rotations and virtual simulation programs should be developed to enhance the quality of the mental health nursing practicum and maximize its effectiveness. Third, standardized guidelines and safe environments should be established during public health infectious disease crises to ensure that these calamities do not limit students’ opportunities to learn skills and develop clinical competencies through the mental health nursing practicum.

## Figures and Tables

**Figure 1 healthcare-12-00685-f001:**
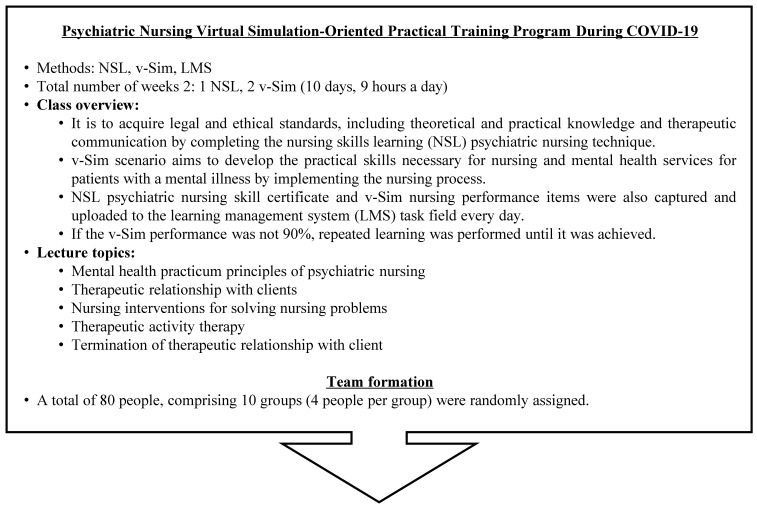
Virtual simulation-centered mental health nursing practicum.

**Figure 2 healthcare-12-00685-f002:**
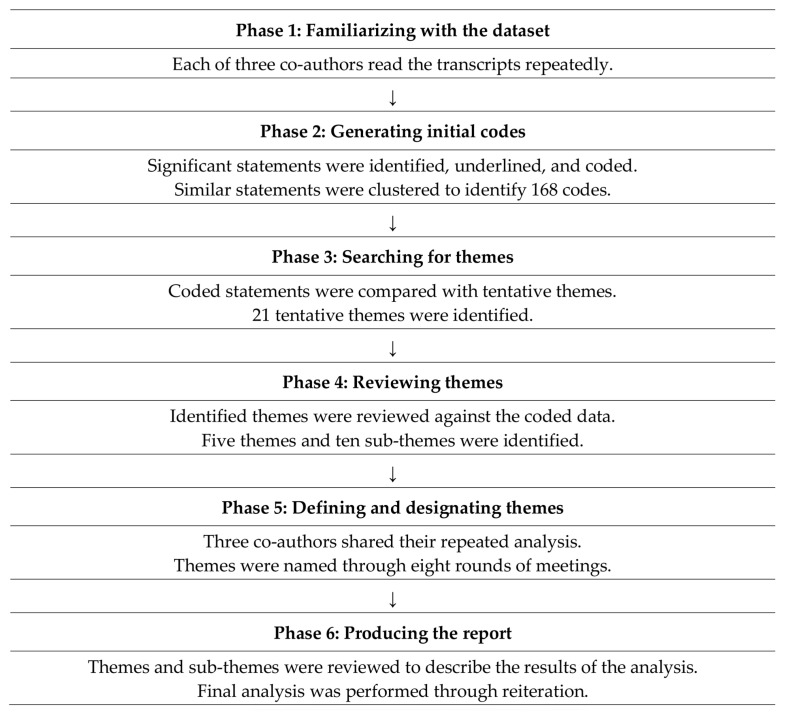
Thematic analysis phases presented by Braun and Clarke (2006).

**Figure 3 healthcare-12-00685-f003:**
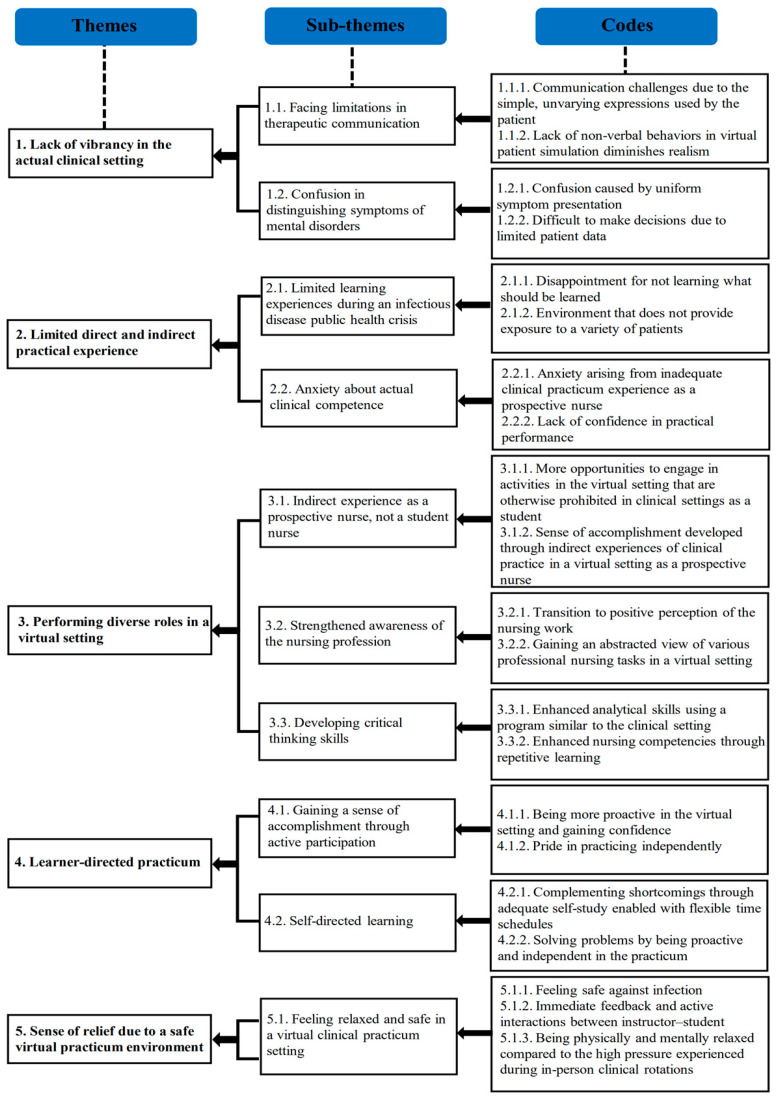
Virtual simulation experience that replaced clinical practice in mental health nursing: themes, sub-themes, and codes.

**Table 1 healthcare-12-00685-t001:** Characteristics of study participants.

Participant	Gender	Age (Years)
1	Woman	22
2	Woman	23
3	Woman	24
4	Woman	23
5	Man	25
6	Man	26
7	Woman	23
8	Woman	26
9	Woman	29
10	Woman	24

**Table 2 healthcare-12-00685-t002:** Key interview questions.

Please tell me about your experience with the virtual simulation program as a replacement for the mental health nursing practicum due to COVID-19. How is the virtual simulation program different from the conventional mental health nursing rotation? What was more helpful about the virtual simulation program compared to the conventional mental health nursing rotation? What was challenging during the virtual simulation program compared to the conventional mental health nursing rotation?

## Data Availability

Data are contained within the article.

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
