# Peer review of "Korean Nursing Students’ Experiences of Virtual Simulation Programs Replacing In-Person Mental Health Nursing Practicum during the COVID-19 Pandemic"

_healthcare, 2024, doi:10.3390/healthcare12060685_

Round 1

Reviewer 1 Report

Comments and Suggestions for Authors

 Qualitative invited article under the title "Korean nursing students' experience with virtual simulation; Programs replacing face-to-face mental health nursing. Practices during the COVID-19 pandemic".

Well stated objective, it seeks to understand and describe nursing students' experiences with virtual simulation programs as a substitute for face-to-face mental health nursing practices during the COVID-19 pandemic.

The objective is met throughout the work, with internal coherence between the theoretical framework presented, the research methodology, the results and the conclusions.

For the theoretical framework, (it is recommended to define clinical simulation) it aims at describing the interactive and simulation tools used by the sample during their simulation practices

- vSim.simulation; reasoning and decision making

- learning (LMS),

- Mosby's Nursing Skills (NSL), which comprises a framework of nursing competencies.

We are talking about skill-based programmes on specific clinical practices and decision-making in the face of problems, ignoring other realities of clinical practice in a mental health context that are essential to guarantee adequate patient care and the humanisation of care; key elements in all areas of care but which are decisive in mental health because they condition the possible therapeutic relationship with the patient, which is essential in quality care.

It should be remembered that each scenario has a complete learning objective (two /3 at the most, but never more; as many as there are realities of the work to be aborted).

In this sense, I miss the reflection by the authors on the design of the simulation scenarios, which is entirely in line with the results of the study; there are other formulas more suitable for this environment; simulated patients (volunteers who behave realistically to simulate clinical interaction, recommended for communication skills, taking the patient's history, asking for informed consent, explaining a procedure, giving bad news); gamification aimed at E-learning; combination of simulations.

Simulation is useful in learning when it is organised for specific learning objectives; in mental health, specific scenarios should be designed to work on communication and relationships with the patient; dramatisation scenarios (psychomotor reactions, delusions, aggression, anxiety and panic), ethical-functional dilemmas (immobilisation due to risk of hetero- or self-harm), situations of harassment of the professional, etc. learn useful strategies on how to deal with it. Each simulation environment aims to teach specific circumstances, and must be adjusted to them. The mental health care that calls for these issues goes far beyond the clinical guideline. And only this practice helps to reduce students' anxiety and stress and develop their communication skills by listening to patients and answering their questions correctly; the foundation of the therapeutic relationship.

Students benefit from role-play scenarios of ethical dilemmas or nurse-to-nurse bullying to learn useful strategies on how to deal with it. Nursing simulation can also help students develop leadership and teamwork skills, which enhances their ability to collaborate and work effectively as a team in complex clinical situations.

Students benefit from role-play scenarios of ethical dilemmas or nurse-to-nurse bullying to learn useful strategies on how to address it. Nursing simulation can also help students develop leadership and teamwork skills, which enhances their ability to collaborate and work effectively as a team in complex clinical situations.

I raise this reflection because I think it should be included in the conclusions, which are certainly very accurate, but obviously based on the simulation programme applied and the type of care required in mental health. And if teamwork between differentiated professionals is essential in any healthcare reality, where it is most evident is in situations where you cannot count on the collaboration of the patient; something more than usual in mental health settings. Coinciding with their findings that simulation is a complementary element to professional training but never a basis for it or a substitute for clinical practice with real patients. The technique (care plan) that is acquired through stimulation cannot replace the history of care, the feeling, the bonds of care. Dialogue with the patient's family or immediate environment (case history, previous experiences, behavioural analysis, pharmacological treatment, patterns giving an overview of the patient.

Functional patterns: (follow-up of the case over time, previous care, socio-occupational insertion, admissions outpatient treatment ; physical situation psychomotor agitation, anguish, psychosis,; need for restraint (immobility, mechanical restraint); disturbed sleep, whether or not it is mediated; make the therapeutic alliance impossible; situations that generate anxiety, identification of effective emotional support; how the situation generated by the covid affects the patient (no exits, no contact, interaction with other people) Real experience with the patient outside the therapeutic reality, so that the care (humanisation of care) is better.

Situations that increase the real workload of the nurse (difficulty in building trust, fluctuations, less collaborative, more psychotic part more present, generates frustration in the professional) and call for organisation and coordination of care in the work team (being aware of several people given the unpredictability, or not knowing how they are going to react...).

The conclusions are logical

Methodology;

Correct and adequate; the use of COREQ is appreciated.  Selective sampling (purposive sampling or targeted sampling) is adequate, taking into account the object of research; they are purposively selected to meet the specified criteria which I have doubts whether two or three (clarify this).

- Students who have done face-to-face practice in mental health prior to covid

- Students who have done clinical practicum with simulation during covid

- DUDA: voluntariness when participating in the study. It is not clear to me if the voluntariness to participate in the study was or was not a requirement.

There are 10 students who meet the two/three requirements; it is surprising that there is no gender parity (especially considering the feminisation of the profession; 2 women and 8 men).

Well structured and coherent interview, perfect analysis of results and adequate verbatim notes. Huge thanks for the figures summarising the whole instrument as well as the codes and subcodes. Impeccable,

In spite of the limitations found as a result of the use of simulation (absence of interpersonal communication and consequences in the nurse-patient relationship and at the therapeutic level, and individualism given the conditions of self-learning, not encouraging teamwork) highlight the great usefulness of positive aspects;

- How important it is for the nurse or student to follow a case from beginning to end something that simulation allows but not clinical practice in a real environment 344; Feeling of accomplishment does everything 371

- Identification with the profession; really feeling what nursing is all about. One participant stated that their 386 professional nursing identity was reinforced 381

- Fostering skills in getting information from patients in a timely manner and managing patient cases. 403

- A sense of relief from a safe virtual practice environment. It is striking that when talking about safety, given the characteristics of the COVID situation, only the risk of contagion (ARI) is mentioned and not direct care (real insecurity, stress and fear of conflictive care in disruptive people).

Conclusions consistent with the results and discussion

- Incorporate the unique culture of mental health nursing and facilitate the learning of therapeutic communication skills (612) and teamwork.

- An emotional approach to care is lacking.

- Clarify that a simulation does not work for everything; each setting works on specific things, never all the things that a professional experiences and faces in a real environment.

- The authors miss the opportunity to claim practices in a simulation environment to work on this culture; emotion, fear, anxiety, uncertainty, stress, difficulty..., elements with which they live in day-to-day life. "It's not all about knowing how to put in a catheter or give a drug".

Publication is recommended with some adjustment (annotate)

Author Response

Response to Reviewer 1 Comments

My co-authors and I wish to re-submit our revised manuscript titled “Korean nursing students’ experiences of virtual simulation programs replacing in-person mental health nursing practicum during the COVID-19 pandemic” with changes that thoroughly address your comments.

We thank you for your thoughtful suggestions and insights. The manuscript has benefited from your thorough feedback. We look forward to working with you to move this manuscript closer to publication in Healthcare.

The manuscript has been revised, and the appropriate changes have been made based on your suggestions. The responses to all comments have been prepared and are given below.

We appreciate your consideration.

Review 1 Comments:  

Point 1: Well stated objective, it seeks to understand and describe nursing students' experiences with virtual simulation programs as a substitute for face-to-face mental health nursing practices during the COVID-19 pandemic.

The objective is met throughout the work, with internal coherence between the theoretical framework presented, the research methodology, the results and the conclusions.

Response 1: Thank you for your thorough review and positive feedback.

Point 2: For the theoretical framework, (it is recommended to define clinical simulation) it aims at describing the interactive and simulation tools used by the sample during their simulation practices.

Response 2: Thank you for providing your detailed review and feedback. Clinical simulation is an educational method that enables learners to experience real-world scenarios in a controlled environment to achieve outcomes similar to those in clinical field practice (Lane & Mitchell, 2013; Waxman, 2010). In this study, we have prioritized nursing behaviors for nursing students using the framework illustrated in Figure 1. To enhance learners’ clinical skills through repeated and self-directed learning, we utilized vSim, an interactive communication tool for instructors and learners, that was translated into Korean, as well as NSL, which assesses the attainment of mental nursing skills goals. Additionally, we have included references supporting the definition of clinical simulation for your review.

2.3. Virtual Simulation-Based Practicum Versus In-Person Mental Health Nursing Practicum

In the virtual simulation-centered mental health nursing practicum, the ten students aimed to meet daily learning goals by completing content on mental health nursing skills based on NSL and engaging in ten mental health nursing scenarios on vSim per the protocol. Students continued their learning until they reached a 90% completion rate, at which point they captured and submitted a screenshot of the score, indicating this achievement to the LMS. Subsequently, group debriefing sessions were held, where students presented their application of the nursing process to scenarios from vSim, and the faculty provided feedback and additional instructions to enhance the sampled students' practical competencies (Figure 1). (lines 134-142)

[Reference] Lane, A.J.; Mitchell, C.G. Using a train-the-trainer model to prepare educators for simulation instruction. J. Cont. Educ. Nurs. 2013, 44, 313-317.

[Reference] Waxman, K.T. The development of evidence-based clinical simulation scenarios: guidelines for nurse educators. J. Nurs. Educ. 2010, 49, 29-35.

Point 3: We are talking about skill-based programmes on specific clinical practices and decision-making in the face of problems, ignoring other realities of clinical practice in a mental health context that are essential to guarantee adequate patient care and the humanisation of care; key elements in all areas of care but which are decisive in mental health because they condition the possible therapeutic relationship with the patient, which is essential in quality care.

Response 3: Thank you for providing your professional opinion on clinical simulation. We have incorporated your insights into the limitations of this study and have discussed directions for future research in the manuscript. Your comments, which include alternatives that can better reflect the psychiatric reality of the clinical field, have been duly acknowledged and addressed in the revised discussion section. We appreciate your valuable feedback.

  1. Discussion

To address these issues, virtual reality-based practicums may be considered in situations that are restricted in clinical practicum, such as an infectious disease public health crisis. Previous findings should be considered, wherein students who engaged in virtual reality-based mental health nursing practicum that vividly resembled the clinical setting of communications and interactions developed increased confidence in therapeutic communication with clients [28]. However, rather than solely focusing on skill-based programs, it is crucial to contemplate the psychiatric field specifically. The care and humanization of individuals with mental disorders must incorporate comprehensive insights into the psychiatric clinical practice field. Therefore, when considering the implementation of mental nursing practice based on virtual reality, actively designing the practice using standardized patients and role-plays can enhance therapeutic communication, interaction, and teamwork—essential components in the field of mental nursing. This approach aims to bridge the gap between virtual reality and the reality of the clinical field, which may be lacking in certain aspects of mental health representation. (lines 539-552)

Point 4: It should be remembered that each scenario has a complete learning objective (two /3 at the most, but never more; as many as there are realities of the work to be aborted).

Response 4: Thank you very much for your insightful feedback regarding the experience of learning. As you rightly pointed out, each scenario in our study has a specific learning goal. The decision to set the achievement rate of nursing in the vSim scenario of the clinical practice replacement program at 90% was driven by the intention to enhance nursing practice competency by addressing the limitations of clinical practice and fostering self-directed, repetitive learning.

Establishing a self-directed learning environment resulted in positive research outcomes, with learners reporting a sense of accomplishment and cultivating critical thinking skills. Moving forward, we acknowledge your advice and will consider practical considerations in the clinical field, aiming to stay within two-thirds of the vSim performance goal achievement rate. Thank you once again for your valuable opinion.

Point 5: In this sense, I miss the reflection by the authors on the design of the simulation scenarios, which is entirely in line with the results of the study; there are other formulas more suitable for this environment;

Response 5: Thank you for your meticulous review and insightful advice. As highlighted in your comment, the vSim virtual practice program had limitations in communication with patients with mental disorders. To address this, we included role-play scenarios in the class structure to enhance teamwork and facilitate communication with these patients. It is crucial to note that, during the study period, face-to-face and clinical practice were entirely prohibited in the country due to the COVID-19 pandemic, leading us to conduct online classes for vSim, LMS, and NSL.

Given the heightened anxiety and fear surrounding COVID-19 and interactions with patients' families and those around them, we experience real-world challenges. We politely request for your understanding as Korea’s cultural and national policy characteristics shaped the mental nursing clinical practice. However, to incorporate your valuable insights, we have included a reflection on our simulation scenario design in the limitations and discussions, aligning with other revision recommendations. This adjustment aims to provide more detailed insights into the design for future studies. We appreciate your feedback, and the manuscript has been revised accordingly.

  1. Discussion

To address these issues, virtual reality-based practicums may be considered in situations that are restricted in clinical practicum, such as an infectious disease public health crisis. Previous findings should be considered, wherein students who engaged in virtual reality-based mental health nursing practicum that vividly resembled the clinical setting of communications and interactions developed increased confidence in therapeutic communication with clients [28]. However, rather than solely focusing on skill-based programs, it is crucial to contemplate the psychiatric field specifically. The care and humanization of individuals with mental disorders must incorporate comprehensive insights into the psychiatric clinical practice field. Therefore, when considering the implementation of mental nursing practice based on virtual reality, actively designing the practice using standardized patients and role-plays can enhance therapeutic communication, interaction, and teamwork—essential components in the field of mental nursing. This approach aims to bridge the gap between virtual reality and the reality of the clinical field, which may be lacking in certain aspects of mental health representation. (lines 539-552)

As an alternative, role-plays or standardized patients can be used to enhance communication skills and interview techniques that are required in mental health nursing practicums, specifically during times when clinical practicum experiences are limited. These approaches could reduce students’ tension and anxiety while boosting their confidence in clinical practice [10,32–34]. Therefore, role-play clinical simulations should be considered to develop effective coping strategies, communication skills, and teamwork skills among learners. (lines 570–576)

Point 6: Simulation is useful in learning when it is organised for specific learning objectives; in mental health, specific scenarios should be designed to work on communication and relationships with the patient; dramatisation scenarios (psychomotor reactions, delusions, aggression, anxiety and panic), ethical-functional dilemmas (immobilisation due to risk of hetero- or self-harm), situations of harassment of the professional, etc. learn useful strategies on how to deal with it. Each simulation environment aims to teach specific circumstances, and must be adjusted to them. The mental health care that calls for these issues goes far beyond the clinical guideline. And only this practice helps to reduce students' anxiety and stress and develop their communication skills by listening to patients and answering their questions correctly; the foundation of the therapeutic relationship.

Response 6: Firstly, thank you for your in-depth advice on simulation. Since your advice and opinion are consistent with Point 5 above, we added both of these items together in the limitations section of the Discussion. Despite the national policy's prohibition of face-to-face classes at the time of the study, we considered virtual simulation limitations, standardized patients, patient history taking, and patient consent to enhance realism. However, practical implementation faced challenges such as school guidelines and budgets.

Unfortunately, the current study cannot be revised to include the time after participants graduated. However, we respectfully accept your comment, and your suggestions have been added to the Discussion as a factor to consider in future research. The Discussion has also been revised in response to this, maintaining consistency with Point 5 above.

  1. Discussion

To address these issues, virtual reality-based practicums may be considered in situations that are restricted in clinical practicum, such as an infectious disease public health crisis. Previous findings should be considered, wherein students who engaged in virtual reality-based mental health nursing practicum that vividly resembled the clinical setting of communications and interactions developed increased confidence in therapeutic communication with clients [28]. However, rather than solely focusing on skill-based programs, it is crucial to contemplate the psychiatric field specifically. The care and humanization of individuals with mental disorders must incorporate comprehensive insights into the psychiatric clinical practice field. Therefore, when considering the implementation of mental nursing practice based on virtual reality, actively designing the practice using standardized patients and role-plays can enhance therapeutic communication, interaction, and teamwork—essential components in the field of mental nursing. This approach aims to bridge the gap between virtual reality and the reality of the clinical field, which may be lacking in certain aspects of mental health representation. (lines 539-552)

As an alternative, role-plays or standardized patients can be used to enhance communication skills and interview techniques that are required in mental health nursing practicums, specifically during times when clinical practicum experiences are limited. These approaches could reduce students’ tension and anxiety while boosting their confidence in clinical practice [10,32–34]. Therefore, role-play clinical simulations should be considered to develop effective coping strategies, communication skills, and teamwork skills among learners. (lines 570–576)

Point 7: Students benefit from role-play scenarios of ethical dilemmas or nurse-to-nurse bullying to learn useful strategies on how to deal with it. Nursing simulation can also help students develop leadership and teamwork skills, which enhances their ability to collaborate and work effectively as a team in complex clinical situations.

Response7: Thank you for your valuable opinion. We revised the Discussion section to reflect this point in a consistent way to the comments made in Points 5 and 6 above.

  1. Discussion

To address these issues, virtual reality-based practicums may be considered in situations that are restricted in clinical practicum, such as an infectious disease public health crisis. Previous findings should be considered, wherein students who engaged in virtual reality-based mental health nursing practicum that vividly resembled the clinical setting of communications and interactions developed increased confidence in therapeutic communication with clients [28]. However, rather than solely focusing on skill-based programs, it is crucial to contemplate the psychiatric field specifically. The care and humanization of individuals with mental disorders must incorporate comprehensive insights into the psychiatric clinical practice field. Therefore, when considering the implementation of mental nursing practice based on virtual reality, actively designing the practice using standardized patients and role-plays can enhance therapeutic communication, interaction, and teamwork—essential components in the field of mental nursing. This approach aims to bridge the gap between virtual reality and the reality of the clinical field, which may be lacking in certain aspects of mental health representation. (lines 539-552)

As an alternative, role-plays or standardized patients can be used to enhance communication skills and interview techniques that are required in mental health nursing practicums, specifically during times when clinical practicum experiences are limited. These approaches could reduce students’ tension and anxiety while boosting their confidence in clinical practice [10,32–34]. Therefore, role-play clinical simulations should be considered to develop effective coping strategies, communication skills, and teamwork skills among learners. (lines 570–576)

Point 8: I raise this reflection because I think it should be included in the conclusions, which are certainly very accurate, but obviously based on the simulation programme applied and the type of care required in mental health. And if teamwork between differentiated professionals is essential in any healthcare reality, where it is most evident is in situations where you cannot count on the collaboration of the patient; something more than usual in mental health settings. Coinciding with their findings that simulation is a complementary element to professional training but never a basis for it or a substitute for clinical practice with real patients. The technique (care plan) that is acquired through stimulation cannot replace the history of care, the feeling, the bonds of care. Dialogue with the patient's family or immediate environment (case history, previous experiences, behavioural analysis, pharmacological treatment, patterns giving an overview of the patient.

Functional patterns: (follow-up of the case over time, previous care, socio-occupational insertion, admissions outpatient treatment ; physical situation psychomotor agitation, anguish, psychosis,; need for restraint (immobility, mechanical restraint); disturbed sleep, whether or not it is mediated; make the therapeutic alliance impossible; situations that generate anxiety, identification of effective emotional support; how the situation generated by the covid affects the patient (no exits, no contact, interaction with other people) Real experience with the patient outside the therapeutic reality, so that the care (humanisation of care) is better.

Situations that increase the real workload of the nurse (difficulty in building trust, fluctuations, less collaborative, more psychotic part more present, generates frustration in the professional) and call for organisation and coordination of care in the work team (being aware of several people given the unpredictability, or not knowing how they are going to react...).

Response 8:  Once again, thank you for your professional advice on simulation. We have revised the paper by adding this concept to the Discussion in a way that is consistent with Points 5, 6, and 7 above.

  1. Discussion

To address these issues, virtual reality-based practicums may be considered in situations that are restricted in clinical practicum, such as an infectious disease public health crisis. Previous findings should be considered, wherein students who engaged in virtual reality-based mental health nursing practicum that vividly resembled the clinical setting of communications and interactions developed increased confidence in therapeutic communication with clients [28]. However, rather than solely focusing on skill-based programs, it is crucial to contemplate the psychiatric field specifically. The care and humanization of individuals with mental disorders must incorporate comprehensive insights into the psychiatric clinical practice field. Therefore, when considering the implementation of mental nursing practice based on virtual reality, actively designing the practice using standardized patients and role-plays can enhance therapeutic communication, interaction, and teamwork—essential components in the field of mental nursing. This approach aims to bridge the gap between virtual reality and the reality of the clinical field, which may be lacking in certain aspects of mental health representation. (lines 539-552)

As an alternative, role-plays or standardized patients can be used to enhance communication skills and interview techniques that are required in mental health nursing practicums, specifically during times when clinical practicum experiences are limited. These approaches could reduce students’ tension and anxiety while boosting their confidence in clinical practice [10,32–34]. Therefore, role-play clinical simulations should be considered to develop effective coping strategies, communication skills, and teamwork skills among learners. (lines 570–576)

Point 9: The conclusions are logical

Response 9: Thank you for your positive feedback.

Point 10: Methodology; Correct and adequate; the use of COREQ is appreciated.  

Response 10: We made an effort to meet the COREQ checklist requirements. Thank you for your positive feedback.

Point 11: Selective sampling (purposive sampling or targeted sampling) is adequate, taking into account the object of research; they are purposively selected to meet the specified criteria which I have doubts whether two or three (clarify this).

- Students who have done face-to-face practice in mental health prior to covid

- Students who have done clinical practicum with simulation during covid

- DUDA: voluntariness when participating in the study. It is not clear to me if the voluntariness to participate in the study was or was not a requirement.

Response 11: Thank you for your feedback on the selection of participants. The study participants were fourth year nursing students who had undergone in-person mental nursing practicum in the clinical field before COVID-19 during their third grade. We included participants whose experience could be compared to their previous clinical practice while engaging in mental nursing practice in that year after the COVID-19 pandemic.

The recruitment announcement was disseminated through posters on social network service (SNS) chat rooms and department bulletin boards. Subsequent interviews were conducted with nursing students who expressed their willingness to participate in the study after completing the consent form. Based on the specified selection criteria, students who confirmed their intention to participate were then selected as participants.

We appreciate your valuable input, and based on your suggestions, we have revised the participant recruitment process. We welcome your feedback on the adjustments that were made.

2.2 Study Participants

The purposive sampling method was used to select participants from a pool of nursing students who had experienced both in-person mental health nursing practicum prior to the COVID-19 pandemic and virtual simulation programs following the outbreak of the COVID-19 pandemic. Participants were selected after they expressed voluntary interest in response to the recruitment announcement and they were found to meet the aforementioned selection criteria. From among the 80 fourth-year nursing students of South Korea’s J University in C Province who had completed a two-week (nine hours per day for a total of ten days) virtual simulation program-based practicum for the course “Mental Health Nursing Practicum 2” in the first semester of 2020 and had prior experience with in-person mental health nursing practicum before the outbreak of the COVID-19 pandemic, ten students were enrolled. Their eligibility was verified, and after obtaining informed consent, these ten participants (eight females and two males) were interviewed until data saturation was reached. None of the participants withdrew their consent or discontinued study participation. (lines 101-114)

Point 12: There are 10 students who meet the two/three requirements; it is surprising that there is no gender parity (especially considering the feminisation of the profession; 2 women and 8 men).

Response 12: As indicated in Table 1, the proportion of participants includes eight women and two men. It is important to note that the gender ratio of the nursing college in this study was 20% for men and 80% for women. Additionally, we have modified the term “Woman” to “Female” in Table 1 for consistency. Your consideration and feedback are highly appreciated.

Point 13: Well structured and coherent interview, perfect analysis of results and adequate verbatim notes. Huge thanks for the figures summarising the whole instrument as well as the codes and subcodes. Impeccable,

In spite of the limitations found as a result of the use of simulation (absence of interpersonal communication and consequences in the nurse-patient relationship and at the therapeutic level, and individualism given the conditions of self-learning, not encouraging teamwork) highlight the great usefulness of positive aspects;

- How important it is for the nurse or student to follow a case from beginning to end something that simulation allows but not clinical practice in a real environment 344; Feeling of accomplishment does everything 371

- Identification with the profession; really feeling what nursing is all about. One participant stated that their 386 professional nursing identity was reinforced 381

- Fostering skills in getting information from patients in a timely manner and managing patient cases. 403

- A sense of relief from a safe virtual practice environment. It is striking that when talking about safety, given the characteristics of the COVID situation, only the risk of contagion (ARI) is mentioned and not direct care (real insecurity, stress and fear of conflictive care in disruptive people).

Response 13: Thank you for your positive feedback and your careful review of the research results.

Point 14: Conclusions consistent with the results and discussion

- Incorporate the unique culture of mental health nursing and facilitate the learning of therapeutic communication skills (612) and teamwork.

- An emotional approach to care is lacking.

- Clarify that a simulation does not work for everything; each setting works on specific things, never all the things that a professional experiences and faces in a real environment.

- The authors miss the opportunity to claim practices in a simulation environment to work on this culture; emotion, fear, anxiety, uncertainty, stress, difficulty..., elements with which they live in day-to-day life. "It's not all about knowing how to put in a catheter or give a drug".

Response 14: First, thank you for your positive feedback on the conclusion. Additionally, we appreciate your careful review and advice. In this second practice, our study participants compared non-face-to-face practice before and after the COVID-19 pandemic, when the challenges faced in the first practice were mitigated. Simulation practices for various subjects, including psychiatric nursing, were conducted, which lessened the overall burden.

We incorporated the emotional aspect of Korean culture, particularly the sensitivity to the risk of infection, which resulted in a positive response and a sense of relief from COVID-19. We respectfully request that you interpret the participants’ negative emotions in the context of their concerns about performing well as prospective nurses in non-face-to-face mental nursing clinical practice (lines 335-343, 553-564).

Your feedback has been invaluable in refining our discussion, and we trust these clarifications enhance the alignment of our conclusions with the study's results.

  1. Discussion

In addition, the virtual simulation practicum has created a psychologically comfortable environment, enhancing learner engagement more than traditional clinical practicums [11]. Nursing students experiencing their initial psychiatric nursing clinical practicum often encountered prejudice, fear, and anxiety about mental illness [39]. However, the study participants demonstrated reduced levels of uncertainty, fear, and anxiety during their second psychiatric nursing practice in the virtual simulation setting. The possible reason may be the emotional aspect of Korean culture, particularly the heightened sensitivity to the risk of infection, resulting in a positive response and a sense of relief from COVID-19. (lines 632-640).

Point 15: Publication is recommended with some adjustment (annotate).

Response 15: Thank you for your positive feedback and specific advice to improve the paper's quality. We have carefully considered your valuable opinions and made the necessary adjustments for publication.

Reviewer 2 Report

Comments and Suggestions for Authors

This study has nursing significance in the future in virtual simulation in psychiatric nursing practice, where therapeutic communication and interaction with clients are essential in infectious disease pandemic situations such as COVID-19.

Methodology and research results 1) Methodological procedures: (1) The advantage is that the methodological procedures were followed through the COREQ checklist, and the approach was taken by the rigor and truth of the research, which are essential in qualitative research. (2) It is easy to visually check the research analysis process in Figure 2 according to Braun & Clarke's thematic analysis process. 2) Research results: Since the coding tree in Figure 3 is specific, there is no difficulty in confirming the relationship between the topics and subtopics of the qualitative research procedures and results. 3) Nursing educational significance: The entire VR-centered practical education process is presented in Figure 1 so that other researchers can apply it to psychiatric nursing practice.

In the limitations section of the study, "The limitations of this study are as follows. First, considering the transferability of qualitative research, caution must be taken when applying this study to other curricula or practice subjects other than psychiatric nursing practice. Second, "Because data on nursing students' experiences mainly depended on participants' statements through interviews based on their memories after completion of the practicum, various data collection methods such as participant observation can be considered to collect abundant data."

The limitations of this study are described. However, the virtual reality program being used by researchers is used primarily as a web-based self-directed learning program.

Among the non-face-to-face practice programs applied by the researcher, Vsim, which utilizes virtual reality, is one of the programs, and caution is required to ensure that the meaning of this study's results is not overly interpreted as an application experience of virtual reality programs. Please include this part in the discussion and limitations. Additional descriptions of the training content applied in LMS and NSL can help readers understand it.

Author Response

Response to Reviewer 2 Comments

My co-authors and I wish to re-submit our revised manuscript titled “Korean nursing students’ experiences of virtual simulation programs replacing in-person mental health nursing practicum during the COVID-19 pandemic” with changes that thoroughly address your comments.

We thank you for your thoughtful suggestions and insights. The manuscript has benefited from your thorough feedback. We look forward to working with you to move this manuscript closer to publication in Healthcare.

The manuscript has been rechecked and your suggestions have made the necessary changes. The responses to all comments have been prepared and given below.

We appreciate your consideration.

Review 2 Comments: Methodology and research results 

Point 1: This study has nursing significance in the future in virtual simulation in psychiatric nursing practice, where therapeutic communication and interaction with clients are essential in infectious disease pandemic situations such as COVID-19.

Methodology and research results 1) Methodological procedures: (1) The advantage is that the methodological procedures were followed through the COREQ checklist, and the approach was taken by the rigor and truth of the research, which are essential in qualitative research. (2) It is easy to visually check the research analysis process in Figure 2 according to Braun & Clarke's thematic analysis process. 2) Research results: Since the coding tree in Figure 3 is specific, there is no difficulty in confirming the relationship between the topics and subtopics of the qualitative research procedures and results. 3) Nursing educational significance: The entire VR-centered practical education process is presented in Figure 1 so that other researchers can apply it to psychiatric nursing practice.

Response 1: Thank you for your careful review and positive feedback.

Point 2:  In the limitations section of the study, "The limitations of this study are as follows. First, considering the transferability of qualitative research, caution must be taken when applying this study to other curricula or practice subjects other than psychiatric nursing practice. Second, "Because data on nursing students' experiences mainly depended on participants' statements through interviews based on their memories after completion of the practicum, various data collection methods such as participant observation can be considered to collect abundant data."

 The limitations of this study are described. However, the virtual reality program being used by researchers is used primarily as a web-based self-directed learning program.

Among the non-face-to-face practice programs applied by the researcher, Vsim, which utilizes virtual reality, is one of the programs, and caution is required to ensure that the meaning of this study's results is not overly interpreted as an application experience of virtual reality programs. Please include this part in the discussion and limitations. Additional descriptions of the training content applied in LMS and NSL can help readers understand it.

Response 2: Thank you for your positive feedback and for sharing your extensive knowledge of qualitative research. According to your points, this information was added to the limitations of the discussion and revised as follows.

  1. Discussion

Second, vSim, which utilizes virtual reality, is a web-based self-directed learning program that was implemented in this study as a non-face-to-face practice program. Caution must be exercised when interpreting the results to avoid an overly generalized application of vSim as representative of the entire spectrum of virtual reality programs. (lines 647-651).

In addition, we added the following to help readers understand the application of LMS and NSL in the Materials and Methods.

2.3. Virtual Simulation-Based Practicum Versus In-Person Mental Health Nursing Practicum

Additionally, the program utilized the Learning Management System (LMS)—an essential tool for supporting learning in an online environment and facilitating interaction between instructors and students [23]—and Mosby’s Nursing Skills (NSL) comprising a 32-item nursing skills framework that was developed by Mosby and translated into Korean under the supervision of the Korean Society of Nursing Science [23–25]. The LMS played a central role in managing offline curricula, including learner grades, progress tracking, report submission, and attendance, by transitioning them into an online space dedicated to communication and collaboration between instructors and learners [23]. The evaluation occurred upon completion of the NSL's psychiatric nursing skills video, which presented evidence-based nursing skills tailored for each subject and aligned with Korean clinical practice. These materials represent the latest advancements in clinical skills training [25]. (lines 122-133).

Reviewer 3 Report

Comments and Suggestions for Authors

Overall, your article presents an intriguing perspective that has the potential to enhance mental health nursing education. However, several areas require clarification. Firstly, the choice of conducting interviews via text messages is surprising and warrants explanation. Additionally, the necessity for follow-up interviews with four students should be justified. Furthermore, it's unclear why the authors assessed and reported on participants' religious practices, and its relevance is not apparent in the findings or analysis. Moreover, including such information might compromise participant anonymity, which raises ethical concerns. I would suggest removing this information. Additionally, it's important to acknowledge and discuss the small sample size of only ten students as a limitation of this study. Regarding format, certain sections could benefit from conciseness, as some repetition was observed.

Comments on the Quality of English Language

Minor editing is needed for English language aspects, specifically focusing on word choice, clarity, and sentence structures.

Author Response

Response to Reviewer 3 Comments

My co-authors and I wish to re-submit our revised manuscript titled “Korean nursing students’ experiences of virtual simulation programs replacing in-person mental health nursing practicum during the COVID-19 pandemic” with changes that thoroughly address your comments.

We thank you for your thoughtful suggestions and insights. The manuscript has benefited from your thorough feedback. We look forward to working with you to move this manuscript closer to publication in Healthcare.

The manuscript has been rechecked and your suggestions have made the necessary changes. The responses to all comments have been prepared and given below.

We appreciate your consideration.

Review 3 Comments: Several areas require clarification.

Point 1: Overall, your article presents an intriguing perspective that has the potential to enhance mental health nursing education.

Response 1: Thank you for your careful review and positive feedback. We have described the area you commented on below.

Point 2: Firstly, the choice of conducting interviews via text messages is surprising and warrants explanation. Additionally, the necessity for follow-up interviews with four students should be justified.

Response 2: Qualitative researchers adhere to rigorous data collection procedures, employing various media, including photos, sounds, visual media, or digital text messages, to elicit participants' responses (Creswell & Poth, 2016). In our study, text messages were incorporated as a means of data collection. Following the initial face-to-face interview, subsequent interviews were conducted using phones and text messages to gather additional data. The decision to conduct follow-up interviews with four participants was made to ensure saturation was reached and that no new data or content emerged. These additional interviews were conducted via phone and text messages due to the challenges of face-to-face interviews, given the participants’ graduation and relocation. We clarified the expression by changing “required for” to “conducted with.”

2.4. Data Collection

Follow-up interviews were conducted with four participants, and the additional interviews were conducted via phone and text messages due to physical distance. (lines 166-168)

[Reference] Creswell, J.W.; Poth, C.N. (2016). Qualitative Inquiry and Research Design: Choosing among Five Approaches, 4th ed.; Sage publications: Thousand Oaks, California, 2018.

Point 3: Furthermore, it's unclear why the authors assessed and reported on participants' religious practices, and its relevance is not apparent in the findings or analysis. Moreover, including such information might compromise participant anonymity, which raises ethical concerns. I would suggest removing this information.

Response 3: Thank you for your careful review. We deleted the religion-related data and description from the Results section and Table 1.

The mean age was 24.5 years (Table 1). (line 114)

Point 4: Additionally, it's important to acknowledge and discuss the small sample size of only ten students as a limitation of this study.

Response 4: Thank you for your opinion. However, in qualitative studies, sample size is not generally necessary; my understanding is that it is essential to collect data until it reaches saturation with a small sample size according to the small doctrine sampling. Accordingly, we 7mentioned the saturation point in Materials and Methods in lines 168-171. We have presented references for the principle of small doctrine sampling (Holloway & Galvin, 2017; Speziale & Galvin, 2011). We hope that this satisfies your concerns.

[Reference] Holloway, I.; Galvin, K. Qualitative Research in Nursing and Healthcare, 4th ed.; Wiley-Blackwell: Chichester, West Sussex, UK, 2017.

[Reference] Speziale, H.S.; Streubert, H.J.; Carpenter, D.R. Qualitative Research in Nursing: Advancing the Humanistic Imperative, 5th ed.; Lippincott Williams & Wilkins: Philadelphia, US, 2011.

2.4. Data Collection

Data collection and analysis proceeded concurrently until no new concepts and significant data emerged; the first author shared the interview transcripts with two other authors (EP, HY), and the point of saturation was determined after independent review and meetings. (lines 168-171)

Reviewer 4 Report

Comments and Suggestions for Authors

The article was written following the rules of a qualitative study. I have some suggestions regarding the article. These suggestions are marked on the file.

Author Response

Response to Reviewer 4 Comments

My co-authors and I wish to re-submit our revised manuscript titled “Korean nursing students’ experiences of virtual simulation programs replacing in-person mental health nursing practicum during the COVID-19 pandemic” with changes that thoroughly address your comments.

We thank you for your thoughtful suggestions and insights. The manuscript has benefited from your thorough feedback. We look forward to working with you to move this manuscript closer to publication in Healthcare.

The manuscript has been rechecked and your suggestions have made the necessary changes. The responses to all comments have been prepared and given below.

We appreciate your consideration.

Review 4 Comments: The article was written following the rules of a qualitative study. I have some suggestions regarding the article. These suggestions are marked on the file.

Response:  Thank you for your positive feedback. We reviewed the part you commented on in the PDF file and revised it as shown below.

Point 1: Please use the same aims

Response 1: Thank you for your careful review and feedback. Following comments from you and the other reviewers, the aims as presented in the title, abstract, introduction, and study design have been organized for consistency, repetitive expressions have been avoided, and duplicated sentences have been deleted. Please see the changes below.

Title: Korean Nursing Students’ Experiences of Virtual Simulation Programs Replacing In-Person Mental Health Nursing Practicum during the COVID-19 Pandemic

Abstract: This qualitative study explored the experiences of nursing students whose clinical practice in mental health nursing has been substituted with virtual simulation programs due to the COVID-19 pandemic. (lines 9-11)

  1. Introduction

1.1. Study Rationale

Therefore, it is imperative to understand in depth nursing students’ perceptions and experiences with virtual simulation programs in mental health nursing. This study aims to explore the experiences of nursing students with mental health nursing virtual simulation programs as an alternative to clinical practicum. The findings would be useful for providing high-quality alternative mental health practicums in the event of restrictions caused by another infectious disease outbreak. For this purpose, qualitative research is beneficial in understanding and describing the experiences and perspectives of students in virtual simulation programs as an alternative to clinical practicum [20]. This study aims to explore nursing students’ experiences with virtual simulation programs for mental health nursing practicum, to eventually provide foundational data for developing effective clinical practicum alternatives in exceptional circumstances such as the COVID-19 situation.  (lines 80-86)

1.2. Aim

This study aims to explore nursing students’ experiences in virtual simulation practice as an alternative to mental health clinical practicum during the COVID-19 pandemic, (lines 89-90)

  1. Materials and Methods

2.1. Study Design

This qualitative study aims to explore the experiences of nursing students using virtual simulation programs, specifically as an alternative to an in-person mental health nursing practicum during the COVID-19 pandemic. (lines 95-97)

Point 2: why do you give this information. If there is a reason, explain. If not, don't give this information.

Response 2: Thank you for your careful review. We deleted the religion-related data and description from the Result section and Table 1.

The mean age was 24.5 years (Table 1). (line 114)

Point 3: Briefly summarize your conclusion from your study findings.

Response 3: Thank you for your valuable opinion. We briefly presented the results of the study in the Conclusion section.

  1. Conclusion

This study is significant as it provides understanding and illustrates nursing students’ experience with virtual simulation during the COVID-19 pandemic. Nursing students encountered the limitations of therapeutic communication with patients with mental disorders due to the lack of vitality in the virtual simulation programs, but their sense of accomplishment improved through independent nursing performance and repeated learning led by learners in a safe virtual practicum environment. (lines 655-660)
